# A New Perspective on Shampoo's Preconditioner

**Depen Morwani***
Kempner Institute
Harvard University
dmorwani@g.harvard.edu

**Itai Shapira***
SEAS
Harvard University
itaishapira@g.harvard.edu

**Nikhil Vyas***
SEAS
Harvard University
nikhil@g.harvard.edu

**Eran Malach**
Kempner Institute
Harvard University
emalach@g.harvard.edu

**Sham Kakade**
Kempner Institute
Harvard University
sham@seas.harvard.edu

**Lucas Janson**
Department of Statistics
Harvard University
ljanson@g.harvard.edu

## Abstract

Shampoo, a second-order optimization algorithm that uses a Kronecker product preconditioner, has recently received increasing attention from the machine learning community. Despite the increasing popularity of Shampoo, the theoretical foundations of its effectiveness are not well understood. The preconditioner used by Shampoo can be viewed as either an approximation of the Gauss–Newton component of the Hessian or the covariance matrix of the gradients maintained by Adagrad. Our key contribution is providing an explicit and novel connection between the optimal Kronecker product approximation of these matrices and the approximation made by Shampoo. Our connection highlights a subtle but common misconception about Shampoo's approximation. In particular, the square of the approximation used by the Shampoo optimizer is equivalent to a single step of the power iteration algorithm for computing the aforementioned optimal Kronecker product approximation. Across a variety of datasets and architectures we empirically demonstrate that this is close to the optimal Kronecker product approximation. We also study the impact of batch gradients and empirical Fisher on the quality of Hessian approximation. Our findings not only advance the theoretical understanding of Shampoo but also illuminate potential pathways to enhance its practical performance.

## 1 Introduction

Second-order optimization methods offer significant theoretical advantages over first-order approaches, promising faster convergence rates by incorporating curvature information. Recently, these methods have seen success in practical large-scale training of neural networks such as Gemini 1.5 Flash (Gemini Team, 2024) and in the Algoperf benchmark (Dahl et al., 2023; MLCommons, 2024). One of the primary challenges in this field arises from the substantial memory and computational demands of traditional second-order methods, such as Adagrad (with full matrix) (Duchi et al., 2011) and Newton's method. When applying classical techniques, they require storing and inverting a $|P| \times |P|$ dimensional matrix $H$ (either covariance of the gradients for Adagrad or the Gauss–Newton component of the Hessian for Newton's method), where $|P|$ denotes the model parameters. With modern architectures often comprising billions of parameters, this leads to quadratic memory and cubic computational requirements, rendering direct application practically infeasible.

---

*Equal contribution. Randomized Author Ordering.

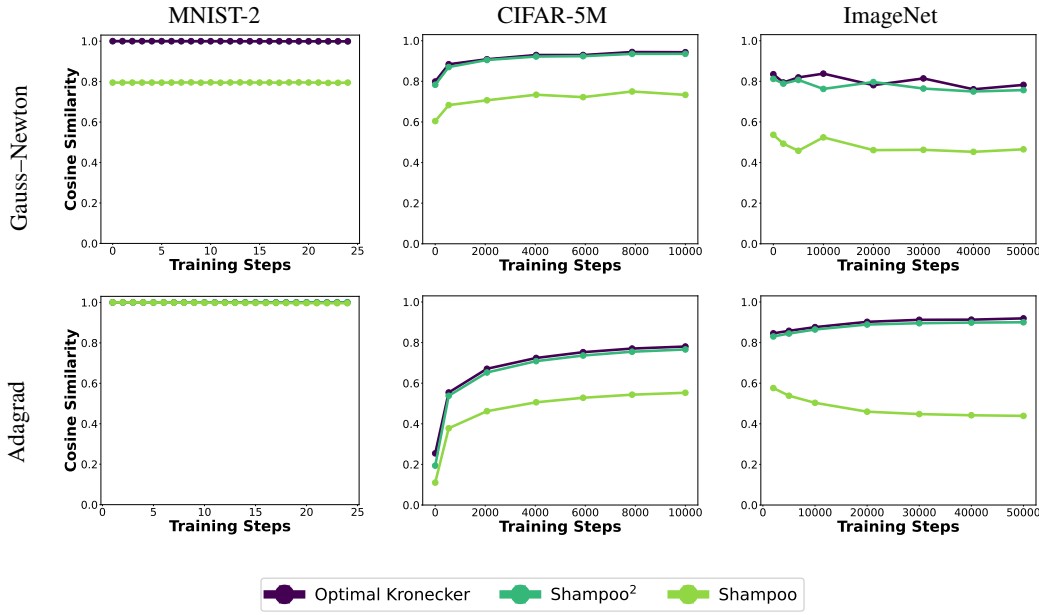

Figure 1: Comparison of Kronecker product approximations for the Gauss–Newton Hessian (top) and Adagrad preconditioner (bottom) across datasets and architectures. Plots show the cosine similarity between the true matrix and three approximations: optimal Kronecker, Shampoo², and Shampoo. Shampoo² closely tracks the optimal Kronecker approximation, outperforming the original Shampoo method, consistent with Proposition 1. For MNIST-2 (binary subsampled), Shampoo² perfectly correlates with the optimal Kronecker approximation as proved in Corollary 2 for binomial logistic regression. See Appendix C for dataset and architecture details.

To address this issue, one line of work focuses on efficient approximations of the matrix $H$ (Gupta et al., 2018; Martens & Grosse, 2015). These methods typically employ either a diagonal approximation (e.g., Adam (Kingma, 2014)) or a layer-wise Kronecker product approximation of $H$. Such approaches are motivated by the significant memory and computational efficiency gains they offer compared to maintaining and inverting the full matrix $H$. Among the most prominent methods utilizing layer-wise Kronecker product approximations are K-FAC (Martens & Grosse, 2015) and Shampoo (Gupta et al., 2018).

In this work, we primarily focus on the Shampoo optimizer (Gupta et al., 2018), which has recently gained increasing attention from the research community. Notably, in the Algoperf benchmark of optimization algorithms proposed for practical neural network training workloads (Dahl et al., 2023), Shampoo appears to outperform all other existing methods. Another recent study, elucidating the Google Ads recommendation search pipeline, revealed that the Google Ads CTR model is trained using the Shampoo optimizer (Anil et al., 2022). Additionally, a recent work (Shi et al., 2023) implemented a distributed data parallel version of Shampoo, demonstrating its superior speed in training ImageNet compared to other methods.

Previous research has introduced the concept of optimal Kronecker product approximation (in Frobenius norm) for a matrix $M$ (Loan & Pitsianis, 1993). This involves finding the projection of $M$ onto the set of matrices expressible as a Kronecker product of two smaller matrices. Loan & Pitsianis (1993) demonstrated that this optimal approximation can be computed numerically using a power iteration scheme. Our work presents a novel result (Proposition 1) that establishes a precise connection between Shampoo's approximation and this optimal Kronecker-factored approximation. Specifically, we show that the *square* of Shampoo's approximation is equivalent to a single step of the power iteration algorithm used to compute the optimal approximation.

Despite its empirical success, the theoretical foundations of Shampoo were not fully understood. Our results bridge this gap by connecting Shampoo to optimal Kronecker approximation of Newton and Adagrad methods. Moreover, our theoretical insights were utilized in the design of SOAP Vyas

et al. (2024), a recently proposed optimizer that improves upon AdamW and Shampoo on language modeling tasks.

The main contributions of the work are summarized below:

- We theoretically show (Proposition 1) that the square of the Shampoo's approximation of $H$ is precisely equal to one round of the power iteration scheme for obtaining the optimal Kronecker factored approximation of the matrix $H$. Informally, for any covariance matrix $H = \mathbb{E}[gg^\top]$ [1] where $g \in \mathbb{R}^{mn}$, we argue that the *right* Kronecker product approximation of $H$ is $\mathbb{E}[GG^\top] \otimes \mathbb{E}[G^\top G]$, while the original Shampoo work (Gupta et al., 2018) proposes $\mathbb{E}[GG^\top]^{1/2} \otimes \mathbb{E}[G^\top G]^{1/2}$, with $G \in \mathbb{R}^{m \times n}$ representing a reshaped $g$ into an $m \times n$ matrix.

- We empirically verify that the result of one round of power iteration (i.e. square of the Shampoo's approximation) is very close to the optimal Kronecker factored approximation (Figure 1), and provide theoretical justification for the same (Section 3.2.1). Our experimental results also show that the approximation proposed by the original Shampoo work (Gupta et al., 2018), which does not use the *square*, is significantly worse.

- For the Hessian based viewpoint of Shampoo (Section 2.2.2), we empirically demonstrate the impact on the Hessian approximation of various practical tricks implemented to make Shampoo more computationally efficient such as averaging gradients over batch (Section 4.1) and using empirical Fisher instead of the actual Fisher (Section 4.2).

**Remark.** It is worth noting that previous works (Balles et al., 2020; Lin et al., 2024) have investigated why Adagrad-based methods such as Adam and Shampoo incorporate an additional square root in their updates compared to the Hessian inverse. While this is an important question, it lies outside the scope of our work. For more details on this topic, we refer readers to Appendix G.

## 2 BACKGROUND

In this section, we provide the technical background, including basic definitions (Section 2.1), two perspectives on Shampoo's optimization (Adagrad in Section 2.2.1, Hessian in Section 2.2.2), and the theory of the optimal Kronecker product approximation (Section 2.3).

### 2.1 NOTATION AND BASIC DEFINITIONS

We use lowercase letters to denote scalars and vectors, and uppercase letters to denote matrices. For a symmetric matrix $A$, $A \geq 0$ (resp. $A > 0$) denotes that $A$ is positive semi-definite (PSD) (resp. positive definite). Similarly, for symmetric matrices $A$ and $B$, $A \geq B$ (resp. $A > B$) means $A - B \geq 0$ (resp. $A - B > 0$). The identity matrix of size $n$ is denoted by $I_n$. We use $M[i, j]$ to refer to the $(i, j)$ entry of the matrix $M$. The Kronecker product of two matrices $A \in \mathbb{R}^{p \times q}$ and $B \in \mathbb{R}^{r \times s}$ is denoted by $A \otimes B \in \mathbb{R}^{pr \times qs}$. It is defined such that $(A \otimes B)[ri + i', sj + j'] = A[i, j]B[i', j']$ where $0 \leq i < p, 0 \leq j < q, 0 \leq i' < r, 0 \leq j' < s$. The vectorization of a matrix $A \in \mathbb{R}^{m \times n}$, denoted by $\mathrm{vec}(A)$, is an $mn$-dimensional column vector obtained by stacking the columns of $A$ on top of one another. We will usually denote $\mathrm{vec}(A)$ by $a$. The Frobenius norm of a matrix $A \in \mathbb{R}^{m \times n}$, denoted by $||A||_F$, is defined as $||A||_F = \sqrt{\sum_{i=1}^{m} \sum_{j=1}^{n} A[i, j]^2}$.

The following is a basic lemma about Kronecker products that will be used later:

**Lemma 1** (Henderson & Searle (1981)). $(A \otimes B) \, \mathrm{vec}(G) = \mathrm{vec}(BGA^\top)$.

### 2.2 SHAMPOO

The Shampoo optimizer, introduced by Gupta et al. (2018), builds on the principles of the Adagrad algorithm (Duchi et al., 2011). Shampoo can be understood through two primary perspectives: as an extension of Adagrad, and as an approximation of the Gauss–Newton component of the Hessian. We explore both perspectives below in Section 2.2.1 and Section 2.2.2 respectively.

---

[1]The Gauss–Newton component of the Hessian can also be expressed as a covariance matrix. For details, refer Section 2.2.2.

### 2.2.1 ADAGRAD BASED PERSPECTIVE OF SHAMPOO

Adagrad is a preconditioned online learning algorithm that uses the accumulated covariance of the gradients as a preconditioner. Given the model parameters $\theta_t \in \mathbb{R}^p$ at iteration $t$, and the gradient $g_t \in \mathbb{R}^p$ of the loss function with respect to $\theta_t$, Adagrad maintains a preconditioner $H_{\text{Ada}} = \sum_{t=1}^{T} g_t g_t^\top$. The parameter update, for a learning rate $\eta$, is given by:

$$\theta_{T+1} = \theta_T - \eta H_{\text{Ada}}^{-1/2} g_T.$$

Shampoo extends Adagrad by maintaining a layer-wise Kronecker product approximation of the full Adagrad preconditioner $H_{\text{Ada}}$. Let $G_t \in \mathbb{R}^{m \times n}$ be the gradient for a weight matrix $W_t \in \mathbb{R}^{m \times n}$ at iteration $t$. The following lemma forms the basis for Shampoo's approximation:

**Lemma 2** (Gupta et al. (2018)). *Assume that $G_1, \ldots, G_T$ are matrices of rank at most $r$. Let $g_t = \text{vec}(G_t)$ denote the vectorization of $G_t$ for all $t$. Then, for any $\epsilon > 0$, we have:*

$$\epsilon I_{mn} + \frac{1}{r} \sum_{t=1}^{T} g_t g_t^\top \preccurlyeq \left( \epsilon I_m + \sum_{t=1}^{T} G_t G_t^\top \right)^{1/2} \otimes \left( \epsilon I_n + \sum_{t=1}^{T} G_t^\top G_t \right)^{1/2}.$$

Building on the above lemma, Shampoo maintains two preconditioners, $L_t \in \mathbb{R}^{m \times m}$ and $R_t \in \mathbb{R}^{n \times n}$, which are initialized as $\epsilon I_m$ and $\epsilon I_n$, respectively. The updates for the preconditioners and the Shampoo parameter update, with learning rate $\eta$, are given by:

$$L_T = L_{T-1} + G_T G_T^\top; \quad R_T = R_{T-1} + G_T^\top G_T; \quad W_{T+1} = W_T - \eta L_T^{-1/4} G_T R_T^{-1/4}.$$

In Lemma 2, Shampoo approximates the Adagrad preconditioner $H_{\text{Ada}} = \sum_{t=1}^{T} g_t g_t^\top$ using the Kronecker product $\left( \sum_{t=1}^{T} G_t G_t^\top \right)^{1/2} \otimes \left( \sum_{t=1}^{T} G_t^\top G_t \right)^{1/2}$. Our main focus is to study the *optimal Kronecker product approximation* of $H_{\text{Ada}}$ and how it relates to Shampoo's approximation.

### 2.2.2 HESSIAN BASED PERSPECTIVE OF SHAMPOO

In this section, we describe the Hessian approximation viewpoint of Shampoo, explored by previous works (Anil et al., 2021; Osawa et al., 2023), as an alternative to the Adagrad-based perspective. Our theoretical and empirical results apply to both viewpoints.

**Gauss–Newton (GN) component of the Hessian.** For a datapoint $(x, y)$, let $f(x)$ denote the output of a neural network and $\mathcal{L}(f(x), y)$ represent the training loss. Let $W \in \mathbb{R}^{m \times n}$ be a weight matrix in the network and let $\mathcal{D}$ denote the training distribution. For cross-entropy (CE) loss, the Gauss-Newton component of the Hessian of the loss with respect to $W$ is given by (see Appendix E for details):

$$H_{\text{GN}} = \mathbb{E}_{(x,y) \sim \mathcal{D}} \left[ \frac{\partial f}{\partial W} \frac{\partial^2 \mathcal{L}}{\partial f^2} \frac{\partial f}{\partial W}^\top \right] = \mathbb{E}_{\substack{x \sim \mathcal{D}_x \\ s \sim f(x)}} \left[ g_{x,s} g_{x,s}^\top \right],$$

Here, $f(x)$ refers to the network's output, and $\mathcal{D}_x$ represents the training distribution of $x$ (Pascanu & Bengio, 2014). The right-hand side is commonly referred to as the Fisher matrix, while its counterpart using real labels, $\mathbb{E}_{(x,y) \sim \mathcal{D}} \left[ g_{x,y} g_{x,y}^\top \right]$, is called the empirical Fisher. Going forward, for simplicity, we will denote the Fisher matrix as $\mathbb{E}_{x,s \sim f(x)} \left[ g_{x,s} g_{x,s}^\top \right]$, asumming $x$ is drawn from $\mathcal{D}_x$ and similarly for both $x$ and $y$ drawn from $\mathcal{D}$.

Optimizers such as K-FAC and Shampoo, when viewed from the Hessian perspective, perform a layerwise Kronecker product approximation of the Fisher matrix $H_{\text{GN}}$. The following lemma establishes Shampoo's approximation:

**Lemma 3** (Adapted from Gupta et al. (2018); Anil et al. (2021)). *Assume that $G_{x,s}$ are matrices of rank at most $r$. Let $g_{x,s} = \text{vec}(G_{x,s})$. Then, for any $\epsilon > 0$,*

$$\mathbb{E}_{x,s \sim f(x)} \left[ g_{x,s} g_{x,s}^\top \right] \preccurlyeq r \left( \mathbb{E}_{x,s \sim f(x)} \left[ G_{x,s} G_{x,s}^\top \right] \right)^{1/2} \otimes \left( \mathbb{E}_{x,s \sim f(x)} \left[ G_{x,s}^\top G_{x,s} \right] \right)^{1/2}. \quad (1)$$

In Lemma 2 the matrix on the left hand side is equal to $H_{\text{GN}}$ and the right hand side represents the Kronecker product approximation made by Shampoo. However, directly computing this approximation at each step is computationally expensive. In practice, Shampoo applies two additional

approximations to make the process more efficient. First, it replaces the per-input gradient by batch gradient, i.e, replacing $\mathbb{E}_{x,s\sim f(x)}[G_{x,s}G_{x,s}^\top]$ with $\mathbb{E}_{B,\mathbf{s}}[G_{B,\mathbf{s}}G_{B,\mathbf{s}}^\top]$, where $B$ denotes a batch of data points, $\mathbf{s}$ is the concatenation of $s \sim f(x)$ for all $(x, y) \in B$ and $G_{B,\mathbf{s}} = \frac{1}{|B|}\sum_{(x,y)\in B, s=\mathbf{s}[x]} G_{x,s}$ is the *sampled batch gradient*, with $\mathbf{s}[x]$ representing the sampled label corresponding to $x \in B$. Second, Shampoo replaces sampled labels with real labels, i.e., it replaces $\mathbb{E}_{B,\mathbf{s}}[G_{B,\mathbf{s}}G_{B,\mathbf{s}}^\top]$ with $\mathbb{E}_B[G_B G_B^\top]$, where $G_B = \frac{1}{|B|}\sum_{(x,y)\in B} G_{x,y}$ is the *batch gradient*.

Thus, if $G_t$ and $W_t$ represent the batch gradient and weight matrix at iteration $t$, and $\lambda$ is an exponential weighting parameter, then the Shampoo update is given by:

$$L_t = \lambda L_{t-1} + (1-\lambda)G_t G_t^\top; \quad R_t = \lambda R_{t-1} + (1-\lambda)G_t^\top G_t; \quad W_{t+1} = W_t - \eta L_t^{-1/4} G_t R_t^{-1/4},$$

where $L_t$ and $R_t$ represent the left and right preconditioners maintained by Shampoo, respectively.

When viewed from the Hessian perspective, our focus is on studying:

- The optimal Kronecker product approximation of the matrix $H_{\mathrm{GN}}$ and its connection to Shampoo's approximation (detailed in Section 3).
- The effect of the two aforementioned approximations (batch gradients and real labels) on the quality of the approximation (detailed in Section 4).

### 2.3 OPTIMAL KRONECKER PRODUCT APPROXIMATION

In this subsection we describe how to find the optimal Kronecker product approximation of a matrix $H \in \mathbb{R}^{mn\times mn}$ under the Frobenius norm. This problem can be reduced to finding the best rank-one approximation of a rearranged version of $H$. We define the rearrangement operator $\mathrm{reshape}()$, applied to a matrix $H$, as follows:

$$\mathrm{reshape}(H)[mi + i', nj + j'] = H[mj + i, mj' + i'],$$

where $i, i' \in [0, 1, \ldots, m-1]$ and $j, j' \in [0, 1, \ldots, n-1]$, and $\mathrm{reshape}(H) \in \mathbb{R}^{m^2\times n^2}$. One useful property of the rearrangement operator is:

$$H = A \otimes B \iff \mathrm{reshape}(H) = ab^\top, \tag{2}$$

where $A \in \mathbb{R}^{m\times m}$, $a = \mathrm{vec}(A) \in \mathbb{R}^{m^2}$, $B \in \mathbb{R}^{n\times n}$ and $b = \mathrm{vec}(B) \in \mathbb{R}^{n^2}$. This property can be used to prove the following result on optimal Kronecker product approximation:

**Lemma 4** (Van Loan & Pitsianis (1993))**.** *Let $H \in \mathbb{R}^{mn\times mn}$ and let $L \in \mathbb{R}^{m\times n}, R \in \mathbb{R}^{n\times m}$. Then, the Kronecker product approximation of $H$ is equivalent to the rank-one approximation of* $\mathrm{reshape}(H)$ *under the Frobenius norm:*

$$\|H - L \otimes R\|_F = \|\mathrm{reshape}(H) - \mathrm{vec}(L)\,\mathrm{vec}(R)^\top\|_F,$$

Since the best rank-one approximation of a matrix is given by its singular value decomposition (SVD), we conclude the following:

**Corollary 1.** *Let $H \in \mathbb{R}^{mn\times mn}$. If the top singular vectors and singular value of $\mathrm{reshape}(H)$ are represented by $u_1, v_1$ and $\sigma_1$, respectively, then the matrices $L \in \mathbb{R}^{m\times m}$ and $R \in \mathbb{R}^{n\times n}$ defined by*

$$\mathrm{vec}(L) = \sigma_1 u_1, \quad \mathrm{vec}(R) = v_1,$$

*minimize the Frobenius norm $\|H - L \otimes R\|_F$.*

**Obtaining SVD by power iteration.** Power iteration is a well-known method for estimating the top eigenvalue of a matrix $M$. It can also be adapted to compute the top singular vectors of a matrix. The iterative procedure for the left singular vector $\ell$ and the right singular vector $r$ is given by

$$\ell_k \leftarrow Mr_{k-1}; \quad r_k \leftarrow M^\top \ell_{k-1}, \tag{3}$$

where $k$ denotes the iteration number.

**Cosine similarity.** We will use cosine similarity between matrices as a measure of approximation quality. For two matrices $M_1$ and $M_2$, it is defined as

$$\frac{\mathrm{Tr}(M_1 M_2^\top)}{\|M_1\|_F \|M_2\|_F}.$$

A cosine similarity value of 1 indicates perfect alignment, while a value of 0 indicates orthogonality.

## 3 OPTIMAL KRONECKER PRODUCT APPROXIMATION AND SHAMPOO

Loan & Pitsianis (1993) describe an approach to find the optimal Kronecker product approximation of a matrix. Koroko et al. (2023) extend this work to derive layer-wise Kronecker product approximations of the Hessian matrix for networks *without weight sharing*. In particular, their Proposition 3.1 relies on the rank-1 structure of gradients for a single sample to efficiently compute $\hat{H}_{\mathrm{GN}}$ in non-weight-sharing networks. Our analysis does not rely on any assumptions on the gradients and hence applies to both weight-sharing and non-weight-sharing networks. While our analysis builds on these works, this is restricted to Section 3.1. Our primary contribution (Section 3.2) lies in establishing a novel connection between the *square* of the Shampoo estimate and the optimal Kronecker approximation.

### 3.1 OPTIMAL KRONECKER PRODUCT APPROXIMATION

This section applies the theory from Section 2.3 to find the optimal Kronecker product approximation of a covariance matrix $H = \mathbb{E}_{g \sim \mathcal{D}_g}[gg^\top]$ for $g \in \mathbb{R}^{mn}$. Both perspectives of Shampoo, described in Section 2.2, focus on Kronecker product approximations of $H$ in the form $L \otimes R$, where $L \in \mathbb{R}^{m \times m}$ and $R \in \mathbb{R}^{n \times n}$, but for different distributions $\mathcal{D}_g$. For the Adagrad viewpoint, $\mathcal{D}_g$ is the uniform distribution over $g_t$, where $t$ refers to the gradient at iteration $t$, giving $H = H_{\mathrm{Ada}}$. For the Hessian viewpoint, $\mathcal{D}_g$ is the distribution over gradients with batch size 1 and sampled labels, leading to $H = H_{\mathrm{GN}}$. To simplify notation, we use $\mathbb{E}[gg^\top]$ to represent $\mathbb{E}_{g \sim \mathcal{D}_g}[gg^\top]$, as our results apply to any distribution $\mathcal{D}_g$. This section explores the optimal Kronecker product approximation for such a generic matrix $H$, examines its connection to Shampoo, and presents experimental validations for $H = H_{\mathrm{Ada}}$ and $H = H_{\mathrm{GN}}$.

Since $g \in \mathbb{R}^{mn}$, each entry of $g$ can be described by a tuple $(i, j) \in [m] \times [n]$. Consequently, each entry of $H$ can be represented by the tuple $((i, j), (i', j'))$. We now introduce the matrix $\hat{H} := \mathrm{reshape}(H) \in \mathbb{R}^{m^2 \times n^2}$, which is a rearrangement of the entries of $H$ (see Section 2). Using Equation (2), we have: $\hat{H} = \mathbb{E}[G \otimes G]$. Furthermore, by Lemma 4, if $L \otimes R$ is the optimal Kronecker product approximation of $H$, then $\ell r^\top$ is the optimal rank-1 approximation of $\hat{H}$, where $\ell = \mathrm{vec}(L)$ and $r = \mathrm{vec}(R)$. Thus, the problem reduces to finding the optimal rank-1 approximation of $\hat{H}$. Applying the power iteration scheme from Equation (3) to estimate the top singular vectors of $\hat{H}$, and using Lemma 1, gives the following updates for the $k$-th step of power iteration:

$$\ell_k \leftarrow \hat{H} r_{k-1} = \mathbb{E}[G \otimes G] r_{k-1} = \mathrm{vec}(\mathbb{E}[G R_{k-1} G^\top]),$$

$$r_k \leftarrow \hat{H}^\top \ell_{k-1} = \mathbb{E}[G \otimes G]^\top \ell_{k-1} = \mathrm{vec}(\mathbb{E}[G^\top L_{k-1} G]).$$

Reshaping the vectors into matrices gives the updates:

$$L_k \leftarrow \mathbb{E}[G R_{k-1} G^\top]; \quad R_k \leftarrow \mathbb{E}[G^\top L_{k-1} G]. \tag{4}$$

### 3.2 ONE ROUND OF POWER ITERATION

Our primary approximation replaces the full power iteration scheme (Equation (4)) with just a single iteration. This leads to the main contribution of our work:

**Proposition 1.** *A single step of power iteration, starting from the identity matrix, for obtaining the optimal Kronecker product approximation of $H$ is exactly equivalent to the square of Shampoo's approximation of $H$.*

*Proof.* The initialization for this single iteration uses the identity matrix, i.e., $I_m$ and $I_n$ for $L$ and $R$, respectively. This reduces the iterative update equations:

$$L_k \leftarrow \mathbb{E}[G R_{k-1} G^\top]; \quad R_k \leftarrow \mathbb{E}[G^\top L_{k-1} G],$$

to the simplified one-step updates:

$$L \leftarrow \mathbb{E}[G G^\top]; \quad R \leftarrow \mathbb{E}[G^\top G].$$

With these expressions for $L$ and $R$, $L \otimes R$ corresponds exactly to the *square* of Shampoo's approximation of $H$ given by the right-hand side of Equation (1). □

As shown in Figure 1 this single step of power iteration closely approximates the optimal Kronecker product approximation for both $H = H_{\text{GN}}$ (top) and $H = H_{\text{Ada}}$ (bottom). In contrast, the upper bound proposed by the original Shampoo work (Gupta et al., 2018) performs significantly worse.

### 3.2.1 WHY INITIALIZE WITH THE IDENTITY MATRIX?

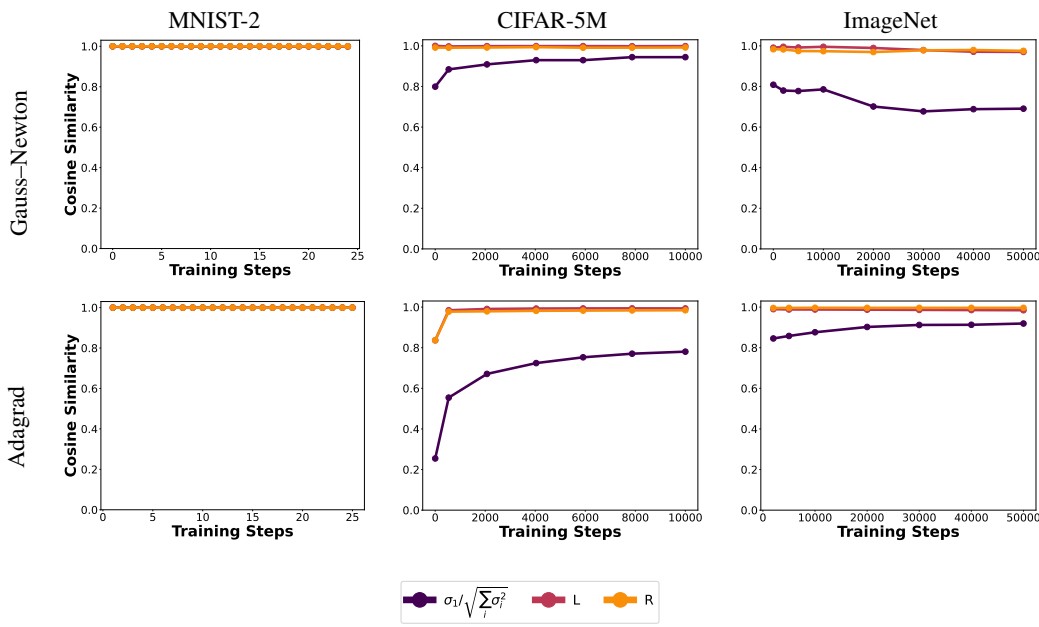

Figure 2: Effectiveness of identity matrix initialization in power iteration for Kronecker product approximation. The plots compare $\frac{\sigma_1}{\sqrt{\sum_i \sigma_i^2}}$, $\frac{\alpha_1 \sigma_1}{\sqrt{\sum_i \alpha_i^2 \sigma_i^2}}$ for left (L) and $\frac{\alpha_1 \sigma_1}{\sqrt{\sum_i \alpha_i^2 \sigma_i^2}}$ for right (R) singular vectors across different datasets and architectures. Top row: Gauss–Newton component of the Hessian ($H_{\text{GN}}$). Bottom row: Adagrad preconditioner matrix ($H_{\text{Ada}}$). Note that $\frac{\alpha_1 \sigma_1}{\sqrt{\sum_i \alpha_i^2 \sigma_i^2}}$ consistently approaches 1 more closely than $\frac{\sigma_1}{\sqrt{\sum_i \sigma_i^2}}$, supporting our theoretical argument for the effectiveness of one-step power iteration with identity initialization. See Appendix C.1 for experimental details.

Suppose the SVD of $\hat{H}$ is given by $\hat{H} = \sum_i \sigma_i u_i v_i^\top$, or equivalently, $H = \sum_i \sigma_i U_i \otimes V_i$. The convergence of the power iteration in one step depends on the inner product of the initialization vector with the top singular vector. Let us focus on the left side,[2] i.e., the update $L \leftarrow \mathbb{E}[GG^\top]$, which as described earlier is equivalent to starting with the initialization $I_n$. Let $\text{vec}(I_n) = \sum_i \alpha_i v_i$, i.e., $I_n = \sum_i \alpha_i V_i$. After one iteration, we obtain $\ell := \sum_i \alpha_i \sigma_i u_i$, and correspondingly, $L := \sum_i \alpha_i \sigma_i U_i$. We are interested in assessing how closely $\ell$ approximates the leading eigenvector $u_1$. The cosine similarity between $\ell$ and $u_1$ is given by $\frac{\alpha_1 \sigma_1}{\sqrt{\sum_i \alpha_i^2 \sigma_i^2}}$.

One reason why the cosine similarity might be large is if $\hat{H}$ is nearly rank-1 (i.e., $\sigma_1$ is large), meaning $H$ is closely approximated by a Kronecker product. However, as shown in Figure 1, this assumption does not universally hold. Instead, we propose an alternative explanation for why a single step of power iteration is typically sufficient: the coefficient $\alpha_1$ is usually larger than $\alpha_i$ for all $i \geqslant 2$. We provide both a theoretical justification and empirical evidence for this.

We start by noting that $\alpha_i = \text{vec}(I_n)^\top v_i = \text{Tr}(V_i)$. Using the identity matrix as initialization is a good choice because it maximizes the dot product with possible top components, i.e., PSD matrices (Proposition 2), and is expected to have a smaller dot product with the later components.

**Lemma 5** ( Loan & Pitsianis (1993)). *$V_1$ is a Positive Semi-Definite (PSD) matrix.*

---

[2]The discussion for the right side is analogous.

Since $V_1$ is a PSD matrix, we want to initialize our power iteration with a matrix that is close to all PSD matrices. We now show that the identity matrix achieves this, specifically by maximizing the minimum dot product across the set of PSD matrices with unit Frobenius norm.

**Proposition 2.** *Consider the set of PSD matrices of unit Frobenius norm of dimension $m$ denoted by $S_m$. Then*

$$\frac{1}{\sqrt{m}} I_m = \arg\max_{M \in S_m} \min_{M' \in S_m} \langle \text{vec}(M), \text{vec}(M') \rangle.$$

This proposition argues that $I_m$ maximizes the worst-case dot product with possible top singular vectors. Now, we argue that its dot product with other singular vectors should be smaller:

**Lemma 6.** *If $V_1$ is positive-definite, then $V_i$ for $i \geqslant 2$ are not PSD.*

Therefore, the diagonal elements of $V_i$ for $i \geqslant 2$ need not be positive, potentially leading to cancellations (for $i \geqslant 2$) in the trace of $V_i$, which equals $\alpha_i$. Hence, we expect $\alpha_i$ for $i \geqslant 2$ to be smaller than $\alpha_1$.

To quantify the benefit of $\alpha_1$ being larger than $\alpha_i$ for $i \geqslant 2$, we compare $\frac{\alpha_1 \sigma_1}{\sqrt{\sum_i \alpha_i^2 \sigma_i^2}}$ (for both left and right singular vectors) and $\frac{\sigma_1}{\sqrt{\sum_i \sigma_i^2}}$. The latter can be interpreted as the cosine similarity if all $\alpha$'s were equal, or as a measure of how close $\hat{H}$ is to being rank-1, since it equals the cosine similarity between $u_1 v_1^\top$ and $\hat{H}$. Thus, $\frac{\sigma_1}{\sqrt{\sum_i \sigma_i^2}}$ corresponds to the "Optimal Kronecker" cosine similarity shown in Figure 1. In Figure 2, we track both quantities during training and observe that $\frac{\alpha_1 \sigma_1}{\sqrt{\sum_i \alpha_i^2 \sigma_i^2}}$ is consistently closer to 1 than $\frac{\sigma_1}{\sqrt{\sum_i \sigma_i^2}}$ for both $H = H_{\text{GN}}$ (top) and $H = H_{\text{Ada}}$ (bottom).

### 3.2.2 EXACT KRONECKER PRODUCT STRUCTURE IN $H$

Our analysis shows that $\mathbb{E}[GG^\top] \otimes \mathbb{E}[G^\top G]$ closely approximates the optimal Kronecker product approximation of $H$. We now show that this approximation becomes exact when $H$ itself is a Kronecker product. In this case, $\hat{H}$ is rank-1, and a single round of power iteration will perfectly recover $\hat{H}$. While our earlier discussion focused on the direction of the top singular vectors of $\hat{H}$, the rank-1 assumption allows us to derive an explicit expression for $\hat{H}$, and consequently for $H$.

**Corollary 2.** *Under the assumption that $\hat{H}$ is rank-1,*

$$H = \left( \mathbb{E}\left[GG^\top\right] \otimes \mathbb{E}\left[G^\top G\right] \right) / \text{Tr}\left( \mathbb{E}\left[GG^\top\right] \right).$$

*Proof.* Let $\hat{H} = \sigma u v^\top$, i.e, $H = \sigma U \otimes V$. Let $I_m = \text{Tr}(U)U + R_m$ and $I_n = \text{Tr}(V)V + R_n$, where $R_m$ and $R_n$ are the residual matrices. After one round of power iteration, the left and right estimates provided by Shampoo are given by $\mathbb{E}\left[GG^\top\right] = \sigma\text{Tr}(V)U$ and $\mathbb{E}\left[G^\top G\right] = \sigma\text{Tr}(U)V$. Hence, we can see that $\text{Tr}\left( \mathbb{E}\left[GG^\top\right] \right) = \sigma\,\text{Tr}(U)\,\text{Tr}(V)$. Thus,

$$H = \sigma U \otimes V = \left( \mathbb{E}\left[GG^\top\right] \otimes \mathbb{E}\left[G^\top G\right] \right) / \text{Tr}\left( \mathbb{E}\left[GG^\top\right] \right).$$

$\square$

Since $H = \hat{H}_{\text{GN}}$ is an $m^2 \times 1$ matrix for binomial logistic regression, it is rank-1, so the equality in the corollary holds. In other words, the square of Shampoo's $H_{\text{GN}}$ estimate perfectly correlates with $H_{\text{GN}}$ for binomial logistic regression. This is demonstrated in the first plot of Figure 1.

We note that $\left( \mathbb{E}\left[GG^\top\right] \otimes \mathbb{E}\left[G^\top G\right] \right) / \text{Tr}\left( \mathbb{E}\left[GG^\top\right] \right)$ as an estimate of Hessian (not Adagrad's covariance matrix) was also derived by Ren & Goldfarb (2021). However, their assumptions were much stronger than ours. Specifically, they assume that the gradients follow a *tensor-normal distribution*, which implies that $\hat{H}$ is rank-1 and that $g$ is mean-zero (thus not applicable to the Adagrad viewpoint). In contrast, our approach only requires a second moment assumption on the gradients: $\hat{H}$ is rank-1. This weaker assumption allows our results to be applicable to a broader range of scenarios, including binomial logistic regression. More importantly, our derivation and experiments show that the *direction* $\mathbb{E}\left[GG^\top\right] \otimes \mathbb{E}\left[G^\top G\right]$ closely approximates the optimal Kronecker product, even if $\hat{H}$ is not rank-1.

### 3.2.3 DISCUSSION ABOUT OPTIMIZATION

Our primary goal in this work was to provide a theoretical foundation for understanding Shampoo's effectiveness, rather than proposing a new algorithm. However, the connection we uncovered between Shampoo$^2$ and the optimal Kronecker product approximation has implications for optimization which we discuss below.

Let us refer to $\mathbb{E}[GG^\top] \otimes \mathbb{E}[G^\top G]$ as $H_1$. As mentioned in Equation (1), Gupta et al. (2018) used the approximation $H_{1/2} \coloneqq \mathbb{E}[GG^\top]^{1/2} \otimes \mathbb{E}[G^\top G]^{1/2}$. In practice, the gradient step in Shampoo is taken in the direction of $H_{1/2}^{-p} \nabla L$, where $p$ is tuned as a hyperparameter (Anil et al., 2021; Shi et al., 2023). Since $H_{1/2}^{-p} = H_1^{-p/2}$, searching over $p$ in $H_{1/2}^{-p}$ yields the same search space as $H_1^{-p}$, meaning that this distinction does not affect optimization speed but deepens our understanding of Shampoo's mechanism.

In fact, empirical work has demonstrated the practical benefits of Shampoo$^2$, where it has shown improved optimization performance as compared to Shampoo (Anil et al., 2021; Shi et al., 2023). Moreover, Vyas et al. (2024) introduced the SOAP algorithm, incorporating our theoretical insights, including Shampoo$^2$ and the trace correction in Section 3.2.2, and showed improvements compared to AdamW and Shampoo.

## 4 HESSIAN APPROXIMATION OF SHAMPOO

In this section, we investigate the impact of different practical considerations on the Hessian approximation in Shampoo, building on the insights from Section 2.2.2. Specifically, we examine the effects of averaging gradients across a batch and using real labels instead of sampled labels. These factors influence how well the Shampoo optimizer approximates the Gauss–Newton matrix $H_{\mathrm{GN}}$, which we previously evaluated with batch size 1 and sampled labels.

### 4.1 AVERAGING GRADIENTS ACROSS THE BATCH

The first factor we analyze is averaging gradients across the batch. We transition from computing the gradient on a per-sample basis:

$$L \leftarrow \mathbb{E}_{x,s \sim f(x)}[G_{x,s}G_{x,s}^\top]; \quad R \leftarrow \mathbb{E}_{x,s \sim f(x)}[G_{x,s}^\top G_{x,s}]$$

to averaging across a batch $B$:

$$L \leftarrow |B|\mathbb{E}_{B,\mathbf{s}}[G_{B,\mathbf{s}}G_{B,\mathbf{s}}^\top]; \quad R \leftarrow |B|\mathbb{E}_{B,\mathbf{s}}[G_{B,\mathbf{s}}^\top G_{B,\mathbf{s}}],$$

where $\mathbf{s}$ denotes the concatenation of $s \sim f(x)$ for all $x \in B$ and $G_{B,\mathbf{s}} = \frac{1}{|B|}\sum_{x \in B, s=\mathbf{s}[x]} G_{x,s}$ is the batch gradient, with $\mathbf{s}[x]$ representing the sampled label for to $x$.

As demonstrated in prior works, this change has no effect in expectation because $G_{x,s}$ is mean-zero for all $x$ when taking the expectation over $s \sim f(x)$ (Bartlett, 1953), i.e. $\mathbb{E}_s[G_{x,s}] = 0$.

**Lemma 7** (Implicitly in Liu et al. (2024); Osawa et al. (2023)).

$$|B|\mathbb{E}_{B,\mathbf{s}}[G_{B,\mathbf{s}}G_{B,\mathbf{s}}^\top] = \mathbb{E}_{x,s \sim f(x)}[G_{x,s}G_{x,s}^\top].$$

This averaging significantly improves running time, providing a multiplicative speed-up proportional to the batch size.

### 4.2 USING REAL LABELS INSTEAD OF SAMPLED LABELS

Next, we consider the impact of replacing sampled labels $s \sim f(x)$ with real labels $y$, as is often done in practice. This shift leads to the empirical Fisher approximation when batch size is 1. Prior work has extensively discussed this approximation and shown that, under certain conditions, the two quantities converge as we move towards optima, in the presence of label noise (Grosse, 2021; Osawa et al., 2023; Kunstner et al., 2019). In Figure 3 (top), we evaluate the approximation of $H_{\mathrm{GN}}$ with batch size 1, finding that the quality remains high throughout training. Yet, as batch size increases, the quality degrades because gradients with real labels are not mean-zero. The following shows how the estimator changes with batch size:

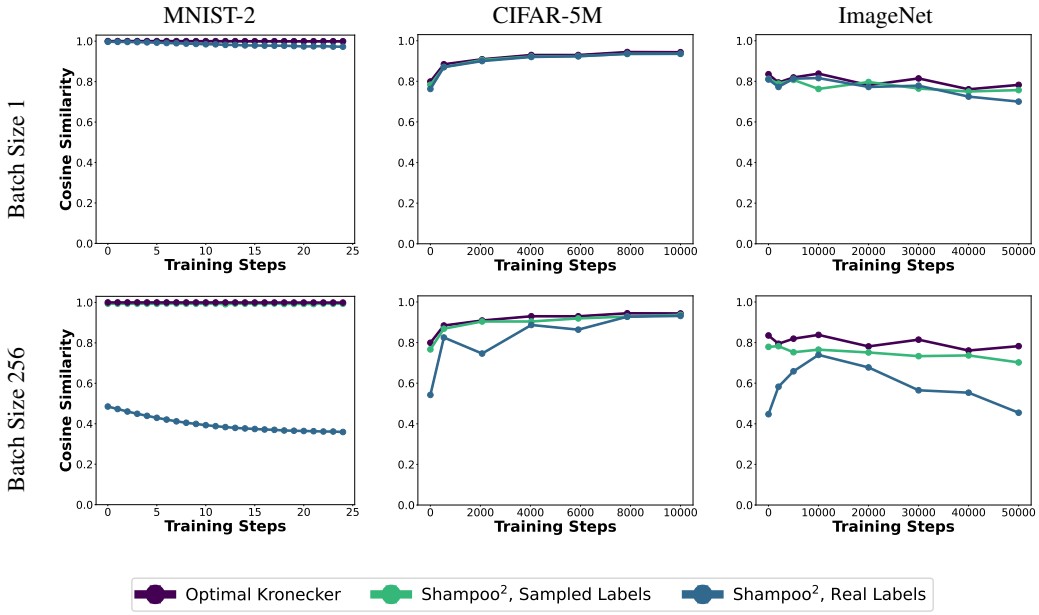

Figure 3: Cosine similarity between approximations of $H_{\mathrm{GN}}$ and the true Hessian. The top row shows results for batch size 1, where the empirical Fisher closely tracks the optimal Kronecker product approximation throughout training. In the bottom row (batch size 256), the approximation quality degrades as batch size increases (see Lemma 8). The batch size refers to that used in the Hessian approximation, not for optimization.

**Lemma 8** (Grosse (2021)). *Let $B$ denote the batch and $G_B = \frac{1}{|B|}\sum_{(x,y)\in B} G_{x,y}$ denote the batch gradient. Then*

$$\mathbb{E}_B[G_B G_B^\top] = \frac{1}{|B|}\mathbb{E}_{x,y}[G_{x,y}G_{x,y}^\top] + \left(1 - \frac{1}{|B|}\right)\mathbb{E}_{x,y}[G_{x,y}]\mathbb{E}_{x,y}[G_{x,y}]^\top.$$

This lemma shows that, depending on the batch size, the estimator interpolates between $\mathbb{E}_{x,y}[G_{x,y}G_{x,y}^\top]$ (the empirical Fisher) and $\mathbb{E}_{x,y}[G_{x,y}]\mathbb{E}_{x,y}[G_{x,y}]^\top$. As shown in Figure 3 (top), at batch size 1, when $\mathbb{E}_B[G_B G_B^\top]$ is equal to $\mathbb{E}_{x,y}[G_{x,y}G_{x,y}^\top]$, it closely tracks the optimal Kronecker product approximation. However, with increasing batch sizes (Figure 3, bottom row), the approximation quality begins to degrade.

We note that this adjustment has the computational advantage of not requiring an additional back-propagation with sampled labels; instead, these computations can be performed alongside standard training.

## 5 CONCLUSION

Our primary contribution is establishing a precise connection between Shampoo's approximation and the optimal Kronecker-factored approximation of matrix $H$. We prove that the square of Shampoo's approximation is equivalent to one round of the power iteration scheme for obtaining this optimal approximation. Empirically, we verify that this single round closely tracks the optimal Kronecker-factored approximation, significantly outperforming the original Shampoo method. Finally, insights from our work have implications for optimization, with recent research showing improvements over AdamW and Shampoo by incorporating our theoretical findings.

ACKNOWLEDGEMENTS

NV and DM are supported by a Simons Investigator Fellowship, NSF grant DMS-2134157, DARPA grant W911NF2010021, and DOE grant DE-SC0022199. This work has been made possible in part by a gift from the Chan Zuckerberg Initiative Foundation to establish the Kempner Institute for the Study of Natural and Artificial Intelligence. SK and DM acknowledge funding from the Office of Naval Research under award N00014-22-1-2377 and the National Science Foundation Grant under award #IIS 2229881. LJ acknowledges funding from the National Science Foundation DMS-2134157.

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

## A    LIMITATIONS

The main contribution of our work is to demonstrate that the square of Shampoo's approximation of $H$ (whether $H$ refers to $H_{\text{Ada}}$ or $H_{\text{GN}}$) is nearly equivalent to the optimal Kronecker approximation of $H$. While we empirically verify this across various datasets and provide theoretical arguments, the gap between the two depends on the problem structure. In some experiments with the ViT architecture (see Appendix B), we observe that the gap is relatively larger compared to other architectures. Furthermore, it remains an open question to understand the conditions—beyond those described in K-FAC (Martens & Grosse, 2015)—under which $H$ is expected to be close to a Kronecker product. Again, in some of the ViT experiments (Appendix B), we find that the optimal Kronecker product approximation to $H$ performs much worse compared to other architectures.

## B    ADDITIONAL EXPERIMENTAL RESULTS

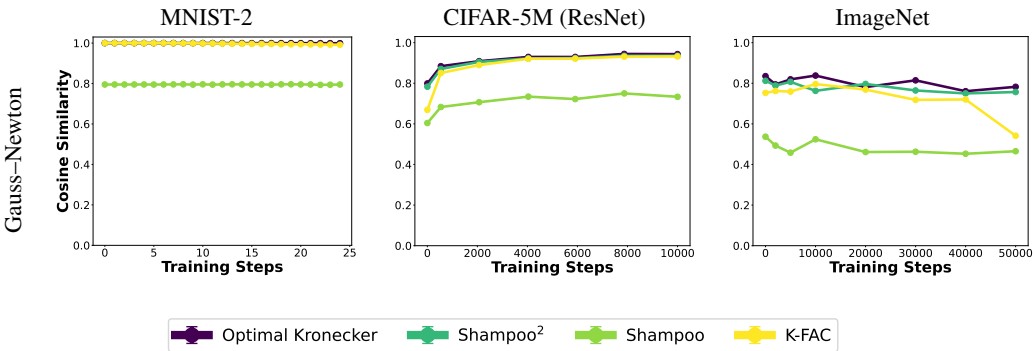

Figure 4: Comparison of different approximations to the Gauss–Newton component of the Hessian. The results show that Shampoo$^2$ achieves higher accuracy in capturing the structure of the true Hessian compared to both Shampoo and K-FAC. K-FAC shows competitive performance in some cases, but Shampoo$^2$ generally offers better approximation.

### B.1    ViT ARCHITECTURE

In this subsection, we present the results for a Vision Transformer (ViT) architecture trained on the CIFAR-5m dataset. This architecture features a patch size of 4, a hidden dimension of 512, an MLP dimension of 512, 6 layers, and 8 attention heads.

For these experiments, we utilize three layers from the fourth transformer block: two layers from the MLP (referred to as 'FFN Linear Layer 1' and 'FFN Linear Layer 2') and the QK layer[3] (referred to as 'Q-K Projection Layer').

## C    EXPERIMENTS

**Datasets and Architectures.** We conducted experiments on three datasets: MNIST (LeCun et al., 1998), CIFAR-5M (Nakkiran et al., 2020), and ImageNet (Deng et al., 2009), using logistic regression, ResNet18 (He et al., 2016), and ConvNeXt-T (Liu et al., 2022) architectures, respectively. For MNIST, we subsampled two digits ($\{0, 1\}$) and trained a binary classifier.

For MNIST, we used the only layer, i.e., the first layer of the linear classifier, for computing the cosine similarities. For Resnet18 and Imagenet, we picked arbitrary layers. In particular, for Resnet18, we used one of the convolution layers within the first block ('layer1.1.conv1' in Resnet18[4]). For Ima-

---

[3]The QK layer is separated from the V part of the layer, following similar decomposition method described by (Duvvuri et al., 2024)

[4]https://pytorch.org/vision/master/_modules/torchvision/models/resnet.html#resnet18

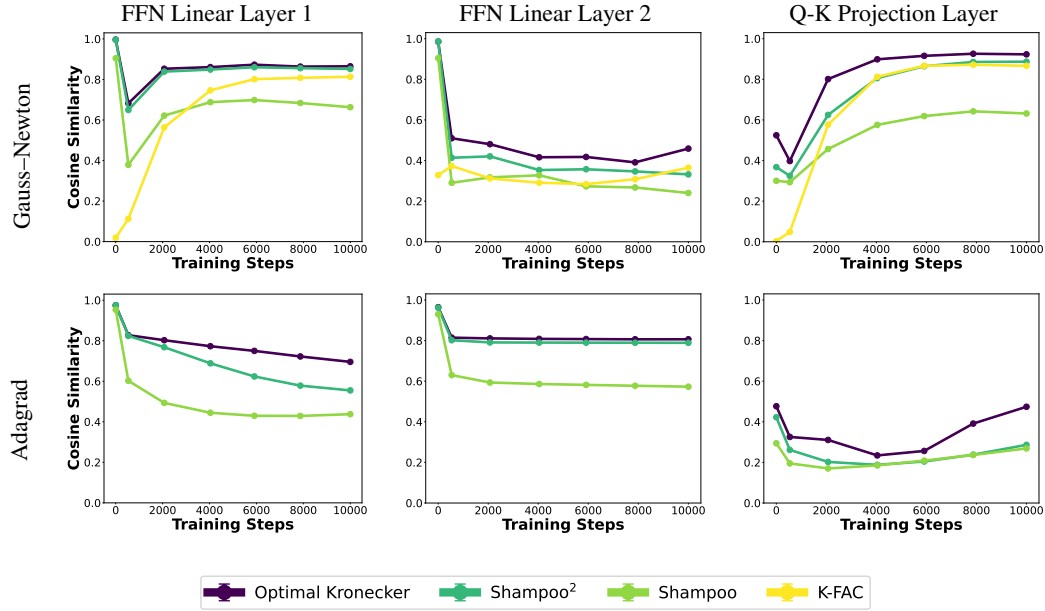

Figure 5: Analogue of Figure 1 for ViT architecture and the CIFAR-5m dataset for 3 layers of the network. For some of the figures we observe relatively larger gaps between Shampoo$^2$ and optimal Kronecker approximation.

Table 1: Summary of Experimental Configurations. $\lambda$ denotes weight decay and $\beta_1$ indicates momentum.

| Dataset | Architecture | Optimizer | Batch Size | Steps | lr | $\lambda$ | $\beta_1$ |
|---|---|---|---|---|---|---|---|
| MNIST | Linear Classifier | GD | Full Batch | 25 | 0.01 | None | 0 |
| CIFAR-5M | ResNet18 | SGD | 128 | 10000 | .02 | None | .9 |
| ImageNet | ConvNeXt-T | AdamW | 2048 | 50000 | 3e-3 | 5e-3 | 0.9 |

genet, we used the 1x1 convolutional layer within the 2nd block of convnext-T ('stages.2.1.pwconv1' in Convnext-T[5]).

**Cosine similarity estimation for $H_{\mathbf{GN}}$.** For estimating the Frobenius norm of $H_{\mathrm{GN}}$, we used the identity:

$$\mathbb{E}_{v\sim\mathcal{N}(0,I_d)}[v^\top H_{\mathrm{GN}}^2 v] = \mathbb{E}_{v\sim\mathcal{N}(0,I_d)}[\|H_{\mathrm{GN}}v\|_2^2] = \|H_{\mathrm{GN}}\|_F^2$$

Hessian-vector products with the Gauss–Newton component were performed using the Deep-NetHessian library provided by Papyan (2019).

For estimating the cosine similarity between $H_{\mathrm{GN}}$ and its estimator $\widetilde{H}_{\mathrm{GN}}$, we used the following procedure:

1. Estimate $\|H_{\mathrm{GN}}\|_F$, and calculate $\|\widetilde{H}_{\mathrm{GN}}\|_F$.

2. Define scaled $\widetilde{H}_{\mathrm{GN}}$ as $\widetilde{S}_{\mathrm{GN}} = \frac{\|H_{\mathrm{GN}}\|_F}{\|\widetilde{H}_{\mathrm{GN}}\|_F} \widetilde{H}_{\mathrm{GN}}$.

3. Cos-sim$(H_{\mathrm{GN}}, \widetilde{H}_{\mathrm{GN}}) = 1 - \frac{\|H_{\mathrm{GN}} - \widetilde{S}_{\mathrm{GN}}\|_F^2}{2\|H_{\mathrm{GN}}\|_F^2}$, where the numerator is again estimated via Hessian-vector products.

---

[5] https://pytorch.org/vision/main/models/generated/torchvision.models.convnext_tiny.html#torchvision.models.convnext_tiny

Note that in the above procedure, we can exactly calculate $\|\tilde{H}_{\mathrm{GN}}\|_F$ as it is generally of a Kronecker product form with both terms of size $m \times m$ or $n \times n$, where $m \times n$ is the size of a weight matrix.

**Cosine similarity estimation for $H_{\mathbf{Ada}}$.** We follow a similar recipe as before, but using a difference method for computing the product $H_{\mathrm{Ada}}v$. For a given time $T$, $H_{\mathrm{Ada}} = \sum_{t=1}^{T} g_t g_t^\top$. Thus, $H_{\mathrm{Ada}}v = \sum_{t=1}^{T}(g_t^\top v)g_t$. We maintain this by keeping a running estimate of the quantity for multiple random vectors $v$ during a training run, and use it for estimating the product $H_{\mathrm{Ada}}v$.

## C.1 FIGURE DETAILS

*Optimal Kronecker* method, wherever used was computed with five rounds of power iteration, starting from the identity. For $H = H_{\mathrm{GN}}$, the Hessian approximations *Shampoo²*, *Shampoo*, and *K-FAC* were done using sampled labels and a batch size of 1. For $H = H_{\mathrm{Ada}}$ and step $t$, we used gradient enocoutered during the training run in steps $\leqslant t$.

*K-FAC* was computed with the "reduce" variant from Eschenhagen et al. (2023).

In Figure 2, the *Optimal Kronecker* legend represents the cosine similarity between the optimal Kronecker approximation of $H_{\mathrm{GN}}$ and $\hat{H}_{\mathrm{GN}}$. This is precisely equal to $\frac{\sigma_1}{\sqrt{\sum_i \sigma_i^2}}$. Similarly, the label $L$ (resp. $R$) represents the cosine similarity between the top left (resp. right) singular vector of $\hat{H}_{\mathrm{GN}}$ and the estimate obtained after one round of power iteration starting from $I_n$ (resp. $I_m$). This is precisely equal to $\frac{\alpha_1 \sigma_1}{\sqrt{\sum_i \alpha_i^2 \sigma_i^2}}$.

In Figure 3 (top), the Hessian approximation is calculated with batch size 1, i.e, $|B| = 1$ in Section 4.2. Similarly, in Figure 3 (bottom), $|B| = 256$.

## D DEFERRED PROOFS

**Lemma 6.** *If $V_1$ is positive-definite, then $V_i$ for $i \geqslant 2$ are not PSD.*

*Proof.* Consider two PSD matrices $M_1$ and $M_2$ having the eigenvalue decomposition $M_1 = \sum \lambda_{1i} q_{1i} q_{1i}^\top$ and $M_2 = \sum \lambda_{2i} q_{2i} q_{2i}^\top$. Then

$$\mathrm{Tr}(M_1 M_2) = \sum_{i,j} \lambda_{1i} \lambda_{2j} \left(q_{1i}^\top q_{2j}\right)^2$$

Thus, if $M_1$ and $M_2$ have unit frobenius norm and $M_1$ is positive definite, then $\mathrm{Tr}(M_1 M_2) > 0$.

Thus, if $V_1$ is positive definite, then by orthogonality of successive singular vectors, $V_i$ for $i \geqslant 2$ cannot be positive semi-definite. $\square$

**Proposition 2.** *Consider the set of PSD matrices of unit Frobenius norm of dimension $m$ denoted by $S_m$. Then*

$$\frac{1}{\sqrt{m}} I_m = \arg\max_{M \in S_m} \min_{M' \in S_m} \langle \mathrm{vec}(M), \mathrm{vec}(M') \rangle.$$

*Proof.* Consider the eigendecomposition of any $M \in S_q$ given by $\sum_{i=1}^{q} \lambda_i v_i v_i^\top$. Denote $L = \{i : \lambda_i \leqslant \frac{1}{\sqrt{q}}\}$. As $\sum \lambda_i^2 = 1$, therefore, $|A| \geqslant 1$. Consider any $j \in A$. Then

$$\langle Vec(M), Vec(v_j v_j^\top) \rangle \leqslant \frac{1}{\sqrt{q}}$$

As $v_j$ is orthogonal to the other eigenvectors. Thus, we can see

$$\max_{M \in S_q} \min_{M' \in S_q} \langle \mathrm{vec}(M), \mathrm{vec}(M') \rangle \leqslant \frac{1}{\sqrt{q}}$$

Moreover, for the matrix $\frac{1}{\sqrt{q}}I_q$, for any matrix $M'$,

$$\frac{1}{\sqrt{q}}\langle I_q, M'\rangle = \frac{\text{tr}(M')}{\sqrt{q}}$$

where $\text{tr}(M')$ denotes the trace of the matrix $M'$. However, we know $\text{tr}(M') = \sum \lambda_i \geqslant 1$ as $\sum \lambda_i^2 = 1$. Thus

$$\frac{1}{\sqrt{q}}\langle I_q, M'\rangle = \frac{\text{tr}(M')}{\sqrt{q}} \geqslant \frac{1}{\sqrt{q}}$$

Note that this is the only matrix with this property as any other matrix will at least have one eigenvalue less than $\frac{1}{\sqrt{q}}$. Thus

$$\frac{1}{\sqrt{q}}I_q = \underset{M \in S_q}{\arg\max}\ \underset{M' \in S_q}{\min}\ \langle \text{vec}(M), \text{vec}(M')\rangle$$

$\square$

**Lemma 7** (Implicitly in Liu et al. (2024); Osawa et al. (2023)).

$$|B|\mathbb{E}_{B,\mathbf{s}}[G_{B,\mathbf{s}}G_{B,\mathbf{s}}^\top] = \mathbb{E}_{x,s\sim f(x)}[G_{x,s}G_{x,s}^\top].$$

*Proof.* Evaluating $G_{B,\mathbf{s}}G_{B,\mathbf{s}}^T$, we get

$$G_{B,\mathbf{s}}G_{B,\mathbf{s}}^T = \frac{1}{|B|^2}\sum_{\substack{x,x'\in B,\\ s=\mathbf{s}[x],s'=\mathbf{s}[x']}} G_{x,s}G_{x',s'}^\top$$

Taking the expectation over $\mathbf{s}$ for a given $B$, and by using $\mathbb{E}_s[G_{x,s}] = 0$ we get

$$\mathbb{E}_{\mathbf{s}}[G_{B,\mathbf{s}}G_{B,\mathbf{s}}^T] = \frac{1}{|B|^2}\sum_x \mathbb{E}_{s\sim f(x)}[G_{x,s}G_{x,s}^\top] = \frac{1}{|B|}\mathbb{E}_{x\sim B,s\sim f(x)}[G_{x,s}G_{x,s}^\top]$$

Now taking an expectation over batches, we get

$$|B|\mathbb{E}_{B,\mathbf{s}}[G_{B,\mathbf{s}}G_{B,\mathbf{s}}^T] = \mathbb{E}_{x,s\sim f(x)}[G_{x,s}G_{x,s}^T]$$

$\square$

**Lemma 8** (Grosse (2021)). *Let $B$ denote the batch and $G_B = \frac{1}{|B|}\sum_{(x,y)\in B} G_{x,y}$ denote the batch gradient. Then*

$$\mathbb{E}_B[G_B G_B^\top] = \frac{1}{|B|}\mathbb{E}_{x,y}[G_{x,y}G_{x,y}^\top] + \left(1 - \frac{1}{|B|}\right)\mathbb{E}_{x,y}[G_{x,y}]\mathbb{E}_{x,y}[G_{x,y}]^\top.$$

*Proof.* Evaluating $G_B G_B^T$, we get

$$G_B G_B^T = \frac{1}{|B|^2}\sum_{(x,y),(x',y')\in B} G_{x,y}G_{x',y'}^\top$$

Taking the expectation over $B$ on both the sides, we get

$$\mathbb{E}_B\left[G_B G_B^T\right] = \frac{1}{|B|^2}\left[|B|\mathbb{E}_{x,y}[G_{x,y}G_{x,y}^\top] + (|B|^2 - |B|)\mathbb{E}_{x,y}[G_{x,y}]\mathbb{E}_{x,y}[G_{x,y}]^\top\right]$$

$$\implies \mathbb{E}_B\left[G_B G_B^T\right] = \frac{1}{|B|}\mathbb{E}_{x,y}[G_{x,y}G_{x,y}^\top] + \left(1 - \frac{1}{|B|}\right)\mathbb{E}_{x,y}[G_{x,y}]\mathbb{E}_{x,y}[G_{x,y}]^\top$$

$\square$

# E    Technical Background on Hessian

**Gauss–Newton (GN) component of the Hessian.** For a datapoint $(x, y)$, let $f(x)$ denote the output of a neural network and $\mathcal{L}(f(x), y)$ represent the training loss. Let $W \in \mathbb{R}^{m \times n}$ represent a weight matrix in the neural network and $\mathcal{D}$ denote the training distribution. Then, the Hessian of the loss with respect to $W$ is given by

$$
\mathbb{E}_{(x,y) \sim \mathcal{D}} \left[ \frac{\partial^2 \mathcal{L}}{\partial W^2} \right] = \mathbb{E}_{(x,y) \sim \mathcal{D}} \left[ \frac{\partial f}{\partial W} \frac{\partial^2 \mathcal{L}}{\partial f^2} \frac{\partial f}{\partial W}^\top \right] + \mathbb{E}_{(x,y) \sim \mathcal{D}} \left[ \frac{\partial \mathcal{L}}{\partial f} \frac{\partial^2 f}{\partial W^2} \right].
$$

The first component, for standard losses like cross-entropy (CE) and mean squared error (MSE), is positive semi-definite and is generally known as the Gauss–Newton (GN) component ($H_{\text{GN}}$). Previous works have shown that this part closely tracks the overall Hessian during neural network training (Sankar et al., 2021), and thus most second-order methods approximate the GN component. Denoting $\frac{\partial \mathcal{L}(f(x), y)}{\partial W}$ by $G_{x,y} \in \mathbb{R}^{m \times n}$ and $g_{x,y} = \text{vec}(G_{x,y})$, for CE loss, it can also be shown that

$$
H_{\text{GN}} = \mathbb{E}_{(x,y) \sim \mathcal{D}} \left[ \frac{\partial f}{\partial W} \frac{\partial^2 \mathcal{L}}{\partial f^2} \frac{\partial f}{\partial W}^\top \right] = \mathbb{E}_{\substack{x \sim \mathcal{D}_x \\ s \sim f(x)}} \left[ g_{x,s} g_{x,s}^\top \right],
$$

# F    Related work

We have already discussed two closely related works Koroko et al. (2023); Ren & Goldfarb (2021) in Sections 3.1 and 3.2.2 respectively. We discuss them again below for completeness.

Ren & Goldfarb (2021) study the Hessian perspective of Shampoo and show that, under the assumption that sampled gradients follow a *tensor-normal* distribution, the square of the Hessian estimate of Shampoo is perfectly correlated with $H_{\text{GN}}$. We also show the same result under much weaker conditions in Corollary 2. Moreover, in Proposition 1 we show that, in general, the square of the Hessian estimate of Shampoo is closely related to the optimal Kronecker product approximation of $H_{\text{GN}}$. We additionally also study the approximations used by Shampoo to make it computationally efficient (Section 4) and the Adagrad perspective of Shampoo's preconditioner.

Loan & Pitsianis (1993) develop the theory of optimal Kronecker product approximation of a matrix (in Frobenius norm). Koroko et al. (2023) use it for finding layer-wise optimal Kronecker product approximation of $H_{\text{GN}}$ for a network without weight sharing. We extend their technique to networks with weight-sharing, and show that the square of the Hessian estimate of Shampoo is nearly equivalent to the optimal Kronecker product approximation of $H_{\text{GN}}$.

Another relevant work is Yao et al. (2021), which introduces AdaHessian, an adaptive second-order optimizer that combines stochastic Hessian diagonal approximations with Adam-style momentum and weighted averaging.

## F.1    Other related works

The literature related to second order optimization within deep learning is very rich, with methods that can be broadly classified as Hessian-free and methods based on estimating the preconditioner $H$ (which could refer to either $H_{\text{Ada}}$ or $H_{\text{GN}}$). Hessian-free methods (Martens, 2010) generally tend to approximate the preconditioned step (for Newton's method) using Hessian vector products, but do not maintain an explicit form of the Hessian. Estimating $H$ (Martens & Grosse, 2015; Gupta et al., 2018) methods maintain an explicit form of the preconditioner that could be efficiently stored as well as estimated.

## F.2    Hessian-free

One of the seminal works related to second order optimization within deep learning was the introduction of Hessian-free optimization (Martens, 2010). The work demonstrated the effectiveness of using conjugate gradient (CG) for approximately solving the Newton step on multiple auto-encoder

and classifications tasks. Multiple works (Martens & Sutskever, 2011; Cho et al., 2015) have extended this algorithm to other architectures such as recurrent networks and multidimensional neural nets. One of the recent works (Garcia et al., 2023) also takes motivation from this line of work, by approximately using single step CG for every update, along with maintaining a closed form for the inverse of the Hessian, for the single step to be effective. Other recent works (Li, 2018; 2024; Pooladzandi & Li, 2024) have focused on designing iterative preconditioners to improve the convergence specifically for stochastic optimization algorithms.

### F.3 ESTIMATING PRECONDITIONER

Given that it is costly to store the entire matrix $H$, various works have tried to estimate layer-wise $H$. KFAC (Martens & Grosse, 2015) was one of the first work, that went beyond diagonal approximation and made a Kronecker product approximation to layer-wise $H_{\text{GN}}$. It showed that this structure approximately captures the per layer Hessian for MLPs. This approximation was extended to convolutional (Osawa et al., 2019) and recurrent (Martens et al., 2018) architectures. Subsequent works also improved the Hessian approximation, by further fixing the trace (Gao et al., 2021) as well as the diagonal estimates (George et al., 2018; Gao et al., 2020) of the approximation. A recent work (Eschenhagen et al., 2023) also demonstrated that K-FAC can be extended to large-scale training.

From the viewpoint of approximating Adagrad (Duchi et al., 2011), Gupta et al. (2018) introduced Shampoo, that also makes a Kronecker product approximation to $H_{\text{Ada}}$. One of the subsequent work (Ren & Goldfarb, 2021) introduced a modification of Shampoo, that was precisely estimating the layer-wise $H_{\text{GN}}$ under certain distributional assumptions. Other works (Anil et al., 2021) introduced a distributed implementation of Shampoo, that has recently shown impressive performance for training large scale networks (Shi et al., 2023). Recently, another paper (Duvvuri et al., 2024) proposed a modification of Shampoo, empirically and theoretically demonstrating that the new estimator approximates $H_{\text{Ada}}$ better than Shampoo's approximation. Our work shows that the square of Shampoo's approximation of $H_{\text{Ada}}$ is nearly equivalent to the optimal Kronecker approximation.

## G COMPARISON WITH EXTRA SQUARE ROOT IN ADAGRAD BASED APPROACHES

Multiple previous works (Balles et al., 2020; Lin et al., 2024) have tried to address the question of why Adagrad-based approaches like Adam and Shampoo, have an extra square root in their update compared to Hessian inverse in their updates. This question is primarily concerned with the final update to the weights being used in the optimization procedure, once we have approximated the Hessian.

The primary contribution of this work is completely orthogonal to this question. We are addressing the question of optimal Kronecker approximation of the Hessian, and its connection to Shampoo's Hessian approximation. This is orthogonal to the Hessian power used in the final update.

