# OpenReview forum: "A New Perspective on Shampoo's Preconditioner"
_ICLR.cc/2025/Conference — ICLR 2025 Poster_

### Official Review · Reviewer_kT8h · 2024-10-29

**Soundness:** 2
**Presentation:** 2
**Contribution:** 1
**Rating:** 3
**Confidence:** 1

**Summary:**

This work provides a theoretical foundation for understanding Shampoo from both Adagrad and Hessian perspective. It also establishes a novel connection between Shampoo’s approximation and the optimal Kronecker product approximation, clarifying misconceptions and highlighting potential improvements.

**Strengths:**

This work analyzes and understands Shampoo from the perspective of Adagrad and Hessian.\
A well-structured work.

**Weaknesses:**

-

**Questions:**

-

---

### Official Review · Reviewer_VKMZ · 2024-11-03

**Soundness:** 3
**Presentation:** 4
**Contribution:** 3
**Rating:** 8
**Confidence:** 4

**Summary:**

Training modern deep neural networks is challenging due to their size and the inherent non-convexity of the training objective.
The most popular optimizers for training these models are first-order methods such as SGD, Adam, and AdamW.
In classical convex optimization, it is well-established empirically and theoretically that methods that incorporate second-order curvature information outperform first-order methods.
However, directly extending these methods to the Deep Learning setting leads to prohibitive computational and memory costs.
Fortunately, more computationally friendly approximate second-order optimizers have emerged in recent years, and they have shown promising performance relative to methods like Adam and AdamW.
Foremost among these methods is Shampoo, which recently won the Algoperf benchmark for training neural networks commonly deployed in practice.

Despite its excellent empirical performance, the reasons underlying Shampoo's success have remained a mystery.
This paper helps explain the method's success by providing a better theoretical understanding of what Shampoo's preconditioner approximates.
The paper provides experiments that corroborate their theory and show it can explain the observed practical performance of variants of Shampoo.
Finally, the paper provides empirical studies investigating how common tricks employed in practice for computational efficiency impact the quality of the Hessian approximation.

**Strengths:**

**Clarity:**

Overall, I found the paper easy and enjoyable to read.
The paper is very well-organized, and the authors make all their points clearly.
For instance, usually, each theoretical result comes with a plot demonstrating the result empirically. This was very nice, made the presentation more concrete, and helped to immediately validate the theory.

**Contribution:**

The theoretical insights into the Shampoo preconditioner are valuable and timely, as Shampoo is arguably the most practically effective approximate second-order method currently available.
This paper's results can possibly lead to improved convergence analyses of Shampoo-like optimizers and inspire the design of new optimizers in this area.
This paper will be of interest to anyone in the ICLR community who is interested in more efficient model training and deep learning optimization.

The experiments in Section 4 give useful practical insights into how computational tricks commonly employed in practice impact the approximation quality of $H_{GN}$.
I think these results can help practitioners to develop better heuristics that balance the tradeoff in the preconditioner delivering better approximation quality vs. computation time.

**Correctness:**

I read through all the proofs of the various results in the paper. They are clear and correct.

**Overall:**

I enjoyed the paper and think it makes a nice contribution to the deep learning optimization literature, so I recommend acceptance.

**Weaknesses:**

**Introduction**

The first paragraph's discussion of the cost of 2nd-order methods should be conditioned a bit; otherwise, it is a bit misleading.
A quadratic space requirement and a cubic computational complexity only arise if you naively try to apply classical techniques like Newton's method to Deep Learning.
By leveraging automatic differentiation, we can compute hvps (which is usually all we need) without forming the Hessian at a cost of $\mathcal O(np)$, where $n$ is the number of samples and $p$ is the number of parameters.
The same holds for the Fisher and Gauss-Newton matrices.
Of course, if no minibatching is done, this too is prohibitively expensive as n and p are both very large.
However, in practice, good performance is often obtained using subsampling.
Many methods in the literature exploit this trick and are called Hessian-free.
The authors are aware of such methods, as they discuss them in the appendix.
Given this, in the final version, I would like the statements of the first paragraph to be qualified.

**Experiments**

One area where the paper could improve is if, in addition to plotting the approximation quality against the iteration count, it also gave plots showing how well the optimizer with that approximation performed on the actual objective loss.
Preferably, these would appear side-by-side.
This would give better insight into the impact of the preconditioner's approximation quality on the optimizer's convergence speed.
For instance, in the setting of Fig 3. it would be interesting to see if $Shampoo^2$-Real Labels exhibited much worse performance than $Shampoo^2$-Sampled Labels, as $Shampoo^2$-Real Labels provides a much worse approximation to $H_{GN}$ in the minibatch setting.

**Related Work**

The authors have been thorough in their list of related work in the appendix, but I feel they should add the citation:

Yao, Z., Gholami, A., Shen, S., Mustafa, M., Keutzer, K. and Mahoney, M., 2021, May. AdaHessian: An adaptive second order optimizer for machine learning. In Proceedings of the AAAI Conference on Artificial Intelligence (Vol. 35, No. 12, pp. 10665-10673).

AdaHessian is another popular stochastic second-order optimizer based on a stochastic approximation to the Hessian diagonal combined with Adam-style momentum and weighted averaging.

**Questions:**

1. In the revision, could you also include plots showing training loss, as per my comment in Weaknesses?
2. Do the authors have any thoughts on why $Shampoo^2$-Real Labels provides a much better approximation quality in the minibatch setting on CIFAR-10 than on ImageNet and Binary MNIST? This is not an essential question; I'm merely curious.

---

> ### Author Response · Authors · 2024-11-17
>
> Thanks for the insightful review. We are happy to see that the reviewer finds our work novel and a useful contribution to the deep learning optimization literature. The specific comments are addressed below:
>
> > The first paragraph's discussion of the cost of 2nd-order methods should be conditioned; otherwise, it is misleading. Quadratic space requirement and cubic computational complexity only arise with naive application of classical techniques.
>
> We agree with the reviewer and will clarify this point in the revision.
>
> > It would be helpful to add side-by-side plots showing both the approximation quality and performance on the objective loss. For example, it would be interesting to see if Real Labels perform worse than Sampled Labels, as Real Labels may provide a worse approximation in the minibatch setting.
>
> Our current focus is on analyzing Shampoo and its variants in relation to Hessian and Adagrad approximations, rather than examining the performance of Real Labels vs Sampled Labels. However, we would be happy to add this plot in the next revision.
>
>
> > The authors should add a citation for AdaHessian: Yao, Z., et al. (2021). AdaHessian: An adaptive second order optimizer for machine learning. AAAI Conference on Artificial Intelligence.
>
> Thank you for pointing this out. We will include the citation in the updated paper.

---

> ### Comment · Reviewer_VKMZ · 2024-11-28
> **Thank you for your reply**
>
> Hi, I apologize to the authors for my late reply. I thank them for their response to comments/questions.
>
> It would be great to include the discussed plot in the final version. I think it should be included, not so much because I'm interested in real labels vs. sampled labels, but do to the fact these two settings exhibit much different approximation quality on some datasets i.e. MNIST-2 and ImageNet. So it would be interesting to see how this impacts training loss, as this shows how approximation quality can directly influence optimization performance.
>
> Overall, the authors have addressed my concerns, and I will maintain my score and recommendation of acceptance.

---

### Official Review · Reviewer_QDot · 2024-11-04

**Soundness:** 3
**Presentation:** 3
**Contribution:** 3
**Rating:** 8
**Confidence:** 3

**Summary:**

The authors analyze the similarity between the optimal Kronecker product approximation and the approximation made by Shampoo and show that the square of the Shampoo preconditioner is equivalent to a single step of the power iteration algorithm for computing the optimal Kronecker product approximation. The authors show that the cosine similarity with H_{Ada} (and Gauss-Newton) of preconditioners over training iterations is very similar for Shampoo^2 and the optimal Kronecker product approximation on MNIST-2, CIFAR-5M and ImageNet. However, this similarity can decay with real labels using a large batch size.

**Strengths:**

Novel insight interpreting a popular optimization algorithm, with extensions to batches and a real data regime. Insights have already been used to create new optimization algorithms which perform well. A good number of numerical experiments clearly verify the claims made about the similarity with the optimal Kronecker product approximation and the reason for choosing L and R as multiples of the identity. The work is succinct with a clear plotting style.

**Weaknesses:**

Experimental details are contained in the Appendix, and we see that cosine similarity changes over the training steps, an investigation into whether this is optimization trajectory independent (or at least using the Shampoo or Shampoo^2 algorithm) could be performed.

**Questions:**

Why was the decision made to compare matrices using cosine similarity? For example, what do the plots look like when the Frobenius norm \| A-B\|_F is used instead? In Figure 3, do you know how Shampoo would perform in the same setting?

---

> ### Author Response · Authors · 2024-11-17
>
> Thanks for the insightful review. We are happy to see that the reviewer finds our work novel and interesting. The specific comments are addressed below:
>
> > Experimental details are contained in the Appendix, and we see that cosine similarity changes over the training steps, an investigation into whether this is optimization trajectory independent (or at least using the Shampoo or Shampoo^2 algorithm) could be performed.
>
> We would be happy to conduct additional experiments to investigate whether the observed changes in cosine similarity are independent of the optimization trajectory if this is critical for the reviewer's assessment of the work.
>
> > Why was the decision made to compare matrices using cosine similarity? What do the plots look like when the Frobenius norm is used instead?
>
>  We chose cosine similarity to maintain consistency with previous work, such as [3]. Additionally, cosine similarity is scale-invariant, and the scale could be absorbed by the learning rate. We are open to including additional comparisons to illustrate the robustness of our results.
>
> [3] Sai Surya Duvvuri, Fnu Devvrit, Rohan Anil, Cho-Jui Hsieh, and Inderjit S Dhillon. Combining axes preconditioners through Kronecker approximation for deep learning. In The Twelfth International Conference on Learning Representations, 2024.
>
> > In Figure 3, do you know how Shampoo would perform in the same setting?
>
> Currently, our experiments focus on Shampoo$^2$ and its connections to Hessian and Adagrad preconditioners. We are happy to conduct additional experiments to compare its performance with Shampoo in the same setting.

---

> > ### Comment · Reviewer_QDot · 2024-11-24
> >
> > Thank you for answering my questions. I am happy to keep my rating the same.

---

### Official Review · Reviewer_sBYH · 2024-11-07

**Soundness:** 3
**Presentation:** 3
**Contribution:** 2
**Rating:** 6
**Confidence:** 3

**Summary:**

The paper presents a theoretical examination of the Shampoo optimizer, a second-order optimization method widely used in machine learning for its efficiency in handling large-scale problems. Shampoo’s preconditioner, based on a Kronecker product, is analyzed in this work to address a gap in understanding its effectiveness. The authors establish that Shampoo's preconditioner approximates the Gauss-Newton matrix or gradient covariance matrix via a novel connection to the power iteration method. Specifically, they show that Shampoo’s approach can be viewed as a single iteration of power iteration aimed at producing an optimal Kronecker product approximation, which they demonstrate empirically to be close to the ideal. Their study further explores the role of batch gradients and the empirical Fisher matrix in improving Hessian approximation, suggesting potential enhancements to Shampoo’s performance in various applications.

**Strengths:**

The authors theoretically and empirically demonstrate that the square of Shampoo’s approximation of $H$ is exactly equivalent to one round of the power iteration method, aimed at obtaining the optimal Kronecker-factored approximation of the matrix $H$. This finding offers a clear theoretical basis for Shampoo’s preconditioner, revealing that its approximation is more rigorous and methodical than previously understood, with direct ties to the power iteration process. This insight deepens our understanding of Shampoo's structure and effectiveness, and it sets the groundwork for further enhancements based on this relationship.

**Weaknesses:**

Despite the fact that I find the results really interesting, I still have my concerns about the relevance of this article to the conference. Below I will try to explain what I mean (if I am wrong, I am open to discussion):

- This paper can hardly be called theoretical, since the authors cite previous work for all lemmas in the paper;

- The whole paper is built on one result (see Proposition 1), which seems to be a mynor result.

I also noticed some typos:

- Line 104: Kronecker product ($\otimes$) is used before definition;
- Line 176 and following: $I_n$ is nowhere defined;
- Line 235: Frobenius norm ($|| \cdot ||_F$) undefined;
- Line 374: PSD is not clear what it means.

**Questions:**

See above

---

> ### Author Response · Authors · 2024-11-17
>
> Thanks for the insightful review. We are happy to see that the reviewer finds our paper very interesting. The specific comments are addressed below:
>
> > This paper can hardly be called theoretical, as the authors cite previous work for all lemmas in the paper.
>
> By “theoretical,” we do not mean that the paper introduces novel proofs or techniques. Rather, our contribution is providing a theoretical explanation of how Shampoo (and Shampoo$^2$) approximate the Hessian and Adagrad preconditioners.  We are happy to revise any language in the paper that may suggest otherwise.
>
> We believe this theoretical understanding is an important contribution to our understanding of the Shampoo optimizer.
>
>
>
> > The whole paper is built on one result (see Proposition 1), which seems to be a minor result.
>
> It is perhaps a minor result in terms of proof complexity, but we believe it makes an important contribution to understanding the Shampoo optimizer. Shampoo and its derivatives (e.g., Distributed Shampoo [1] and SOAP [2]) are among the first optimizers to outperform AdamW in practice. Moreover, subsequent work, such as SOAP, directly builds on the theoretical perspective on Shampoo developed in this paper to further improve the optimizer.
>
>
> [1] Hao-Jun Michael Shi, Tsung-Hsien Lee, Shintaro Iwasaki, Jose Gallego-Posada, Zhijing Li, Kaushik Rangadurai, Dheevatsa Mudigere, and Michael Rabbat. A distributed data-parallel PyTorch implementation of the distributed shampoo optimizer for training neural networks at scale, 2023.
>
> [2] Nikhil Vyas, Depen Morwani, Rosie Zhao, Itai Shapira, David Brandfonbrener, Lucas Janson, and Sham Kakade. SOAP: Improving and stabilizing shampoo using Adam. arXiv preprint arXiv:2409.11321, 2024.
>
>
>
> >I also noticed some typos
>
> Thank you for pointing out these issues. We will address them in the revised paper.

---

> ### Author Response · Authors · 2024-11-23
>
> We would like to thank the reviewer again for their valuable feedback. We would be grateful if the reviewer would consider updating their score based on our response. If not, we would be happy to answer any additional questions.

---

> > ### Comment · Reviewer_sBYH · 2024-11-25
> >
> > Dear Authors,
> >
> > I appreciate the thorough responses, and I'll maintain my current positive score.

---

### Comment · Area_Chair_XE4Y · 2024-11-25
**Important: Please Review Rebuttals and Update Reviews as Needed**

Dear Reviewers,

Thank you for your hard work and dedication to providing thoughtful reviews for this year’s ICLR submissions. Your efforts play a vital role in maintaining the conference’s high standards and fostering meaningful discussions in the community.

As we are close to the end of the discussion phase, I kindly urge you to read the authors’ responses and reevaluate your reviews carefully, especially if they address your concerns. If you find that the authors have resolved the issues or clarified misunderstandings, please consider adjusting your comments and scores accordingly. This ensures fairness and gives each submission the opportunity to be judged on its merits.

Your continued commitment is greatly appreciated—thank you for contributing to the success of ICLR!

---

### Meta-Review · Area_Chair_XE4Y · 2024-12-18

**Metareview:**

The paper presents a compelling and insightful theoretical analysis of the Shampoo optimizer by providing a new perspective on its preconditioner. The authors demonstrate that the preconditioner used in Shampoo can be interpreted as a Kronecker product approximation of the Gauss-Newton or gradient covariance matrix, and this approximation can be understood as a single step of the power iteration method. This novel connection gives a clearer theoretical justification for Shampoo’s performance, helping to address previous misunderstandings regarding its approximation quality. The experiments conducted across different datasets and architectures further validate these claims, showing that the Shampoo preconditioner closely approximates the optimal Kronecker product factorization in practice.

Another strength of the paper is that it goes beyond theory by exploring the impact of practical aspects, such as batch gradients and the empirical Fisher matrix, on the quality of the Hessian approximation. The paper is well-organized and provides clear explanations that are easy to follow, with empirical results that support the theoretical contributions.

**Additional Comments On Reviewer Discussion:**

During the reviewer discussion phase, there was a consensus among the majority of reviewers that the paper makes a valuable contribution to the understanding of second-order optimization methods, particularly the Shampoo optimizer. Regarding Reviewer kT8h’s evaluation, it was noted that the reviewer explicitly mentioned a lack of confidence in assessing the paper. Given this, their rating and feedback were not weighted in the final decision.

---

### Decision · Program_Chairs · 2025-01-22

Accept (Poster)